# Transcriptional Reprogramming of Autographa Californica Multiple Nucleopolyhedrovirus Chitinase and Cathepsin Genes Enhances Virulence

**DOI:** 10.3390/v15020503

**Published:** 2023-02-11

**Authors:** Jeffrey J. Hodgson, A. Lorena Passarelli, Peter J. Krell

**Affiliations:** 1Department of Molecular and Cellular Biology, University of Guelph, Guelph, ON N1G 2W1, Canada; 2Division of Biology, Kansas State University, Manhattan, KS 66502, USA

**Keywords:** baculovirus, enzymes, chitinase, cathepsin, biopesticide

## Abstract

The baculoviral chitinase (CHIA) and cathepsin (V-CATH) enzymes promote terminal insect host liquefaction, which aids viral progeny dissemination. Recombinant Autographa californica nucleopolyhedrovirus (AcMNPV)-derived viruses were previously generated with reprogrammed *chiA* transcription by replacing the native promoter with the AcMNPV *polyhedrin* (*polh*) or core protein (*p6.9*) promoter sequences, but of both these *chiA*-reprogrammed viruses lacked *v-cath* transcription and V-CATH enzymatic activity. Here, we report that dual *p6.9*/*polh* promoter reprogramming of the adjacent *chiA/v-cath* genes resulted in modulated temporal transcription of both genes without impacting infectious budded virus production. These promoter changes increased CHIA and V-CATH enzyme activities in infected *Spodoptera frugiperda*-derived cultured cells and *Trichoplusia ni* larvae. In addition, larvae infected with the dual reprogrammed virus had earlier mortalities and liquefaction. This recombinant baculovirus, lacking exogenous genomic elements and increased *chiA/v-cath* expression levels, may be desirable for and amenable to producing enhanced baculovirus-based biopesticides.

## 1. Introduction

The primary function of the baculovirus chitinase (CHIA) and cathepsin protease (V-CATH) enzymes is to ensure terminal liquefaction of host larval cadavers. Liquefaction of infected insect larvae liberates progeny and aids in progeny virus dissemination [1]. The native baculovirus *chiA* and *v-cath* genes are not conserved in all baculoviruses, and are each considered an auxiliary gene [1,2] for per os infection and the cellular production of budded and occluded virions. It has been suggested that either gene can be transformed into a virulence factor by altering its expression profile [3,4,5]. Overexpression and secretion of AcMNPV [6,7] or insect-derived chitinase [8] increased the virulence (lower LD_50_) and decreased host survival time (shorter ST_50_). AcMNPV overexpression of *Sarcophaga peregrina* ScathL protease also significantly increased virus virulence and decreased host insect feeding [9]. These data indicate that the native viral CHIA and V-CATH enzymes could individually be repurposed as virulence factors by simply altering their expression profiles and/or protein-targeting sites during infection. 

The antiparallel *chiA/v-cath* gene locus (Figure 1a) is conserved in many baculoviruses, particularly group I alphabaculoviruses [10]. AcMNPV *chiA* and *v-cath* are coordinately expressed, and a molecular CHIA-proV-CATH interaction retains both proteins in the endoplasmic reticulum (ER) [11]. Release of both CHIA and proV-CATH from cells and proteolytic maturation of proV-CATH to V-CATH coincides with cell death [12], and V-CATH activation is also linked to nuclear lysis and cellular occlusion body release [13]. The inactive, pro-enzyme form of AcMNPV or Bombyx mori NPV V-CATH (proV-CATH) requires the ER-resident (KDEL-tailed) CHIA as a chaperone for its proper folding, proteolytic activation, co-dependent role in host cadaver liquefaction, and environmental dissemination of progeny viral occlusion bodies [14,15,16]. However, we subsequently determined that when CHIA, lacking its native KDEL, is expressed, it allows premature CHIA/proV-CATH release from cells [11]. Further, in the absence of any CHIA protein expression, proV-CATH folds and proteolytically matures into active V-CATH enzyme [17]. Considering the co-dependent roles in regulating their own cellular release [13], V-CATH maturation [12], and host tissue degradation [1,18], it is expected that simultaneous overexpression of both *chiA* and *v-cath* would further increase the lethality of baculovirus infection relative to native AcMNPV, or if either gene is overexpressed individually. 

We reported that the mRNA and protein expression profiles of CHIA, in its native genomic locus, could be reprogrammed by swapping the native late *chiA* promoter with that of the promoter for either the late *p6.9* or very late *polh* genes [19]. The *polh* promoter enabled higher (4.5X) CHIA enzyme production (at 48 h post infection [p.i.]), but the *p6.9* promoter resulted in earlier transcription of *chiA* and increased CHIA enzyme levels (1.5X) at 48 h p.i. in infected cells. These single *chiA*-reprogrammed viruses lacked both *v-cath* transcription and associated V-CATH protease expression. Here, we used the same strategy to generate a dual *chiA/v-cath* reprogrammed virus (Acp6.9-chiA/polh-cath), which co-overexpresses both enzymes. Compared to the control virus (AcMNPV-Rep), Acp6.9-chiA/polh-cath has differential transcription (RNA) patterns of both *chiA* and *v-cath* RNA and protein production levels of CHIA and proV-CATH. Acp6.9-chiA/polh-cath-infected SF-21 cells and *Trichoplusia ni* larvae contain considerably higher CHIA and V-CATH protease activities than wild-type promoter-expressed counterparts. We also compared the pathology of fifth instar *T. ni* larvae infected with Acp6.9-chiA/cath, a *chiA* only transcriptionally reprogrammed virus (Acp6.9-chiA), and a *chiA/v-cath*-null intermediate virus, AcEGFP, used to generate the *chiA/v-cath* reprogrammed viruses. Infection with the dual reprogrammed Acp6.9-chiA/polh-cath virus, in which *chiA* is expressed under the *p6.9* promoter and *v-cath* under the *polh* promoter, led to earlier larval death and liquefaction relative to infection with AcEGFP, Acp6.9-chiA, or AcMNPV-Rep.

## 2. Methods

### 2.1. Cells, Viruses, and Insects

The *Spodoptera frugiperda*-derived SF-21 cells were cultured at 27 °C in Grace’s Insect Medium (Invitrogen), containing 10% fetal bovine serum, penicillin G (60 μg/mL), and streptomycin sulfate (200 μg/mL). The *Spodoptera frugiperda*-derived Sf9 (clonal isolate 9 from cell line IPLB-SF21-AE) cells were cultured at 27 °C in TC-100 medium (Gibco), containing 10% fetal bovine serum (Atlanta Biologicals/R&D Systems, Minneapolis, MN), penicillin G (60 μg/mL), streptomycin sulfate (200 μg/mL), and amphotericin B (0.5 μg/mL). The E2 strain of AcMNPV was the parental strain, from which AcEGFP was derived by homologous recombination, as previously described [19]. The single *chiA*-reprogrammed virus Acp6.9-chiA/v-cath (abbreviated Acp6.9-chiA hereafter), AcMNPV-repaired with its native intergenic *chiA/v-cath* promoter sequence restored (simply called AcMNPV-Rep hereafter), and the dual *chiA/v-cath* reprogrammed virus Acp6.9-chiA/polh-cath, with the *p6.9* promoter driving *chiA* and the *polh* promoter driving *v-cath*, were all derived from AcEGFP by homologous recombination, as described elsewhere [19]. Briefly, the AcMNPV-Rep virus (i.e., a virus in which the native 45 bp intergenic *chiA/v-cath* promoter region was swapped with the *polh-egfp* cassette in AcEGFP) was derived from AcEGFP, using a donor plasmid (pBSK.CCnative; [21]) and methods [19] described earlier. The *polh*-promoter-based *v-cath*-reprogrammed gene construct was isolated as an XbaI/EcoRI fragment from the pBSK.chiA/ph-CA plasmid [17] and cloned into pBSK.p6.9.chiA/v-cath [19] to produce pBSK.p6.9-chiA/polh-cath. The pBSK.p6.9-chiA/polh-cath plasmid was co-transfected with AcEGFP genomic DNA into SF-21 cells, and Acp6.9-chiA/polh-cath was isolated, and the intergenic promoter sequences and genome structures were assessed, as described previously [19]. Budded viruses were replicated and titrated by end-point dilution [22], using SF-21 cells (when used for cell culture experiments) or Sf9 cells (when used for insect experiments).

*T. ni* larvae (3rd instar) were obtained from Benzon Research (Carlisle, PA) and synthetic insect diet was from Southland Products, Inc. (Lake Village, AR). Viruses were constructed in complete (with antibiotics and 10% serum) TC-100 medium. Fifth instar *T. ni* larvae were injected, using a gauge 22 Hamilton syringe, with 10 µL of virus suspensions, containing 5 × 10^6^ TCID_50_ units per µL (total dose of 5 × 10^4^ TCID_50_ units) of each virus and placed in individual containers containing artificial diet and maintained at 27 °C. Control insects were injected with 10 µL of complete TC-100 growth medium lacking virus. Insects were assessed every 8 h for responsiveness to prodding with a blunt glass rod, in order to determine the time of death. To assess insect liquefaction, insects were inspected and photographed every 24 h. Insects were maintained at 27 °C on a weighing dish, set in a sealed plastic container humidified with a damp tissue and were photographed daily. Experiments to assess insect pathology and mortality were repeated three times. Insects that died or pupated within the first 48 h were not included in the studies; about thirty larvae per virus/treatment were analyzed in each replicate. Insects were photographed with an Olympus digital camera, model F4000. Larval survival data were analyzed with GraphPad Prism 9.2.0. 

### 2.2. RNA Isolation, Electrophoresis, and Blotting

Total RNA was isolated from infected SF-21 cells (m.o.i. of 10 PFU/cell) at 0–48 h p.i., using the RNeasy-TRIzol method (Bowtell & Sambrook, 2003). Mock-infected cell RNA was isolated at *t* = 0. High-Range RNA molecular weight ladder (Fermentas, Waltham, MA) and 5 μg of each denatured RNA sample were electrophoresed in denaturing (2.2 M formaldehyde, 1X MOPS) 1.3% agarose gels. Following electrophoresis, RNA was transferred to positively charged nylon membranes, according to the instructions of the manufacturer (Schleicher and Schuell, Keene, NH). 

### 2.3. Northern Blot Hybridization and Analysis

Gel-purified (gel extraction kit, Qiagen, Germantown, MD) *chiA* and *v-cath* PCR amplicons from agarose served as templates in PCRs incorporating only one (nested) primer for amplification of DIG-labelled ssDNA, complementary to either *chiA* or *v-cath* RNA, as described earlier [19]. RNA blots for each virus construct were pre-treated (30 min, 42 °C, 20 mL DIG Easy Hyb) and hybridized in 10 mL hybridization solution (15 h, 42 °C). Probes were labeled using the DIG nucleic acid detection kit and protocol (Roche, Little Falls, NJ) and 1 mL CSPD substrate (Roche) per blot. Chemiluminescence was detected with X-ray film (Kodak X-Omat).

### 2.4. Temporal CHIA and V-CATH Protein Analysis 

SF-21 cells (1 × 10^6^) were infected with AcEGFP, AcMNPV-Rep, Acp6.9-chiA, or Acp6.9-chiA/polh-cath at an m.o.i. of 10 PFU/cell. At the indicated times p.i., total cellular protein was collected by resuspension, concentrated by centrifugation (500× *g*, 5 min), and analyzed by immunoblotting with either anti-chitinase antibody [Bm-CHI-h; [23,24]] or anti-V-CATH antibody [25]. We assessed AcEGFP and mock-infected cells (at 48 h p.i.) as negative controls for either CHIA or proV-CATH detection. E-64 cysteine protease inhibitor was used to supplement (to 20 µM) cell lysis buffer (100 mM NaCl, 25 mM Tris pH 7.5, 0.5% NP-40, 1% Triton-X 100) to block sodium dodecyl sulfate-induced proV-CATH autoactivation into V-CATH [26]. We analyzed duplicate protein samples from cells infected with AcMPNV-Rep, Acp6.9-chiA, or Acp6.9-chiA/polh-cath separated on gels and blotted, so that we could compare AcMNPV-Rep CHIA and proV-CATH expression profiles to those of Acp6.9-chiA and Acp6.9-chiA/polh-cath and to one another (see full blots in Appendix A). Gels and blots processed with the same lysate samples were each assessed using different antibodies to detect either CHIA, proV-CATH, or GAPDH (as a loading control). The rabbit polyclonal anti-V-CATH (which recognizes both proV-CATH and V-CATH) was used at 1:1000 dilution, and the anti-BmCHI-h was used at 1:50,000 dilution. For immunoblot analysis of detergent (0.5% NP-40 & 1% Triton-X)-soluble/insoluble CHIA and proV-CATH in infected SF-21 cells, lysates components were separated into soluble and insoluble fractions by centrifugation (12,000× *g*, 10 min, 4 °C). Equivalent volumes of soluble and insoluble protein samples among the different viruses were loaded for CHIA and V-CATH analysis, but three times the amount of lysate proteins were used for detection of V-CATH. 

To assess protein and enzymes from insects, fifth instar *T. ni* larvae were injected, as described above, and processed at two, three, or four days p.i. Processing commenced with the collection of pooled hemolymph from several (5–7) insects infected by each virus. For this, a hind proleg was cut with scissors, and the hemolymph was dripped into a chilled 1.5 mL microtube on ice. Hemolymph was immediately mixed with 2X loading buffer (4% [*w*/*v*] sodium dodecyl sulfate, 0.2% [**w*/*v**] bromophenol blue, 200 mM β-mercaptoethanol, 100 mM Tris-Cl, pH 6.8), containing 50 μM of the cysteine protease inhibitor E-64 (AG Scientific, San Diego, CA, USA), heated to 95 °C (5 min), and stored at −20 °C until proteins were separated by SDS 12%-PAGE. Cadavers were collected in separate tubes and frozen (−80 °C). Insect cadavers were homogenized by grinding them with a ground glass pestle in a tube in 50 mM sodium phosphate (NaH_2_PO_4_), pH 7.0 (about three insects per 1 mL). Homogenates were clarified by centrifugation twice at 12,000× *g*, and the supernatants were collected and stored frozen (−80 °C). Since insects collected on day four p.i. were of various melanization and liquefaction states, hemolymph was not collected. These insects were instead frozen (−80 °C) in their individual 1-ounce growth containers prior to homogenization. Protein concentrations of (20X diluted) clarified homogenates were determined using the BCA protein assay kit (Pierce Biologicals, Waltham, MA, USA) and diluted in 50 mM NaH_2_PO_4_, pH 7.0 buffer. 

### 2.5. Chitinase and Cathepsin Activity Assays

Chitinase and cathepsin levels in infected SF-21 cells were measured at 48 h p.i. in three independent experiments, as described earlier [19]. Mock- and virus-infected cells (48 h p.i.) were assessed for chitinase activity, using 50 μg total protein and CM-chitin-RBV substrate (Loewe Biochemica GmbH, Sauerlach, Bayern, Germany) [19], and cysteine protease activity, using 400 μg total protein and azocasein substrate, in the presence and absence of a cysteine protease inhibitor (E-64, 20 μM), based on a previously described assay [15]. Enzyme assays of infected insect cadavers were similar but 200 μg or 400 μg of total homogenate protein was used for the CHIA or V-CATH assays, respectively. T-tests (unpaired, two-tailed, 95% CI) were performed using GraphPad Prism 9.2.0. 

## 3. Results

We previously reported on *chiA*-reprogrammed viruses retaining most of the native *v-cath* promoter sequence (26 nucleotides) adjacent to the AcMNPV-derived *chiA*-reprogrammed using *p6.9* or *polh* promoter sequences [19]. In that study, the *p6.9*-driven *chiA* and *polh*-driven *chiA* temporal mRNA transcription profiles were reprogrammed accordingly, based on the promoter used (*p6.9* or *polh*), and the CHIA enzyme activities of each virus were increased at 48 h p.i. However, both of these viruses lacked *v-cath* mRNA transcription and V-CATH protease, since the intergenic promoter region was altered. We developed a selectable parental AcEGFP virus [19], from which we derived the Acp6.9-chiA/v-cath and Acpolh-chiA/v-cath reprogrammed viruses but wanted to verify that AcEGFP-derived viruses (Figure 1a) could still express *v-cath* properly. To this end, we generated an AcMNPV repair virus (AcMNPV-Rep) containing the native *chiA* and *v-cath* intergenic promoter sequence, using AcEGFP as a backbone and a donor plasmid [19,21] (Figure 1a,b). This virus was called AcMNPV-Rep. In AcMNPV-Rep, the *chiA* and *v-cath* mRNA transcript sizes, temporal transcription profiles, and CHIA and V-CATH enzyme production (at 48 h p.i.) were similar to that of AcMNPV [19], indicating that the AcEGFP-derived viruses did not have obvious defects in *chiA* and *v-cath* transcription and subsequent enzymatic activity (Figure 2 and Figure 3). Transcript levels differed depending on the strength of the promoter used (Figure 2a), with *p6.9*- and *polh*-driven genes having stronger transcription. The AcEGFP-derived virus (with native *chiA/v-cath* expression; AcMNPV-Rep) was subsequently used as the control AcMNPV virus in the remainder of the experiments.

### 3.1. Dual Reprogramming of chiA and v-cath Expression

Previous viral recombinant constructs used to study *chiA* lacked *v-cath* [19,27], making it hard to study the collaborative function of these genes on viral pathogenesis. We decided to use two different promoters to overexpress the genes, the late *p6.9* promoter to reprogram *chiA* expression and the very late *polh* promoter to reprogram *v-cath* expression (Figure 1a) in a single dual-reprogrammed virus (Acp6.9-chiA/polh-cath). The *chiA* and *v-cath* are naturally late-expressed genes, but they can be overexpressed from the stronger very late *polh* promoter without drastically hampering virus replication [19,25]. We performed Northern blotting to compare the *chiA* and *v-cath* mRNA expression profiles in the dual reprogrammed virus, Acp6.9-chiA/polh-cath, to that of AcMNPV-Rep, where the genes were driven by their native late promoters (Figure 2a). The native size of the 2.6 kb *chiA* transcript [19] was not affected by using the *p6.9* and *polh* promoters in the *chiA*/*v-cath* intergenic region. The *p6.9* promoter-derived *chiA* mRNAs were consistently more abundant, beginning at 9 h p.i., than those produced from the wild-type *chiA* promoter [19]. The native size of the 1.5 kb *v-cath* transcript [19] was also unaffected by dual *chiA/v-cath* reprogramming. The *polh* promoter-expressed *v-cath* mRNAs were first detected at 9 h p.i., at levels typical of *polh* transcription patterns up to 18 h p.i. [28,29]. At 24 and 48 h p.i., *polh*-mediated *v-cath* transcription rapidly increased to levels exceeding those expressed from the native *v-cath* promoter. This is consistent with previously reported *polh* expression levels [19,28,30]. Altogether, this analysis indicates that the back-to-back reprogramming using *p6.9* (for *chiA*) and *polh* (for *v-cath*) promoters did not interfere with transcription of either gene.

### 3.2. CHIA and proV-CATH Protein Expression Profiles from Acp6.9-chiA/polh-cath

We also determined the temporal CHIA and proV-CATH protein production profiles of AcMNPV-Rep, Acp6.9-chiA, and Acp6.9-chiA/polh-cath by immunoblotting. We did not detect CHIA or proV-CATH in either control (mock- or AcEGFP-infected cells) cell lysates, although GAPDH (a host protein) was detectable in both controls (Figure 2b). CHIA from AcMNPV-Rep was detectable, starting from 16 h p.i., reaching peak levels by 32 h p.i., and then remaining at similar levels until 48 h p.i. (Figure 2a), similar to that described earlier for AcMNPV [19,31]. CHIA from Acp6.9-chiA, like that of AcMNPV-Rep, was detectable starting from 16 h p.i. (Figure 2b). However, *p6.9* promoter-expressed CHIA accumulated to higher levels at 16 h p.i. compared to AcMNPV-Rep. The peak level of Acp6.9-chiA CHIA expression occurred at 24 h p.i., in contrast to 32 h p.i. in AcMNPV-Rep samples. The CHIA expression profile of Acp6.9-chiA/polh-cath was similar to that of Acp6.9-chiA (Figure 2a), starting at 16 h p.i., peaking at 24 h p.i., and continuing at high levels until 48 h p.i. Similar levels of CHIA protein were produced by the Acp6.9-chiA and Acp6.9-chiA/polh-cath, despite differences in *v-cath* transcription (Figure 2a) and protein production (Figure 2b).

We detected proV-CATH starting at 24 h p.i. in AcMNPV-Rep (Figure 2c), which agrees with the timing at which this protein has been detected from AcMNPV in other studies [15,19,25]. ProV-CATH production by AcMNPV-Rep gradually increased throughout the time-course and was most abundant at 48 h p.i. As reported [19], we did not detect proV-CATH from the Acp6.9-chiA samples, since this construct does not express *v-cath*. Acp6.9-chiA/polh-cath produced detectable proV-CATH from 24 h p.i., similar to that of AcMNPV-Rep, but increased for Acp6.9-chiA/polh-cath from 40 to 48 h p.i. This is consistent with the initial *prov-cath* transcription detected in AcMNPV-Rep and Acp6.9-chiA/polh-cath (Figure 2a). There were similar amounts of proV-CATH detected between AcMNPV-Rep and Acp6.9-chiA/polh-cath protein samples up to 32 h p.i., but by 40 h p.i., more proV-CATH was detected in the Acp6.9-chiA/polh-cath sample compared to AcMNPV-Rep (Figure 2c). ProV-CATH production increased after 32 h p.i. for Acp6.9-chiA/polh-cath (Figure 2b), which is consistent with the stronger *polh* promoter and a burst of expression of *v-cath* transcripts detected between 24 and 48 h p.i., compared to AcMNPV-Rep (Figure 2a). Therefore, temporal *chiA* and *v-cath* mRNA expression patterns altered in Acp6.9-chiA/polh-cath (Figure 2a) agree with the increased immunodetection of CHIA and proV-CATH compared to that for AcMNPV-Rep (Figure 2b). 

CHIA and proV-CATH are regulated co-dependently, and they are retained as a soluble CHIA-proV-CATH complex in the ER until virus-induced cell lysis occurs [11]. When Slack et al. (1995) monitored *polh* promoter-overexpressed AcMNPV *v-cath*, they noted considerable amounts of insoluble proV-CATH [25]. Since the expression profiles of both CHIA and proV-CATH were altered in Acp6.9-chiA/polh-cath relative to AcMNPV-Rep, we wanted to determine if, like AcMNPV [11], the CHIA and proV-CATH co-traffic in Acp6.9-chiA/polh-cath-infected cells. To examine this, we used a detergent (NP-40)-based lysis buffer to fractionate and subsequently analyze the distribution (soluble or insoluble) of CHIA and proV-CATH by immunoblotting (Figure 2c). There was some NP-40-insoluble proV-CATH detected in lysates from AcMNPV-Rep- and Acp6.9-chiA/polh-cath-infected cells. Insoluble CHIA was, however, detected only in the Acp6.9-chiA/polh-cath lysate (Figure 2b). Therefore, despite overall abundant expression of *chiA* mRNA (Figure 2a) and similar total CHIA protein (Figure 2b) in Acp6.9-chiA and Acp6.9-chiA/polh-cath, insoluble CHIA was detected only in cells infected with Acp6.9-chiA/polh-cath. CHIA and proV-CATH broadly co-fractionated (in soluble or insoluble fractions), agreeing with the co-fractionation data of other CHIA forms (truncated and epitope-tagged or fluorescent protein-fused) [17], and this is consistent with proV-CATH and CHIA co-trafficking in cells, even when both of their expression profiles are modified. 

### 3.3. Dual Transcriptional Reprogramming Increases CHIA and V-CATH Enzymatic Activity

To assay for activity of total (soluble and insoluble) CHIA and V-CATH, we disrupted cells by sonication and used equal amounts of total protein in chitinase and cathepsin enzymatic assays. We compared CHIA and V-CATH activities between AcMNPV-Rep and the reprogrammed viruses at 48 h p.i. (Figure 3). No chitinase activity was detected from the mock- or AcEGFP-infected control samples (Figure 3a). The CHIA activity of AcMNPV-Rep was less than that of Acp6.9-chiA, agreeing with our previous studies [19]. Acp6.9-chiA/polh-cath had similar CHIA activity to that of Acp6.9-chiA. Both Acp6.9-chiA and Acp6.9-chiA/polh had slightly higher levels than AcMNPV-Rep, which is in agreement with previous work [19]. 

We used an azocasein-based chromogenic protease assay [25] to compare proteolytic activities in the different virus-infected cell homogenates (Figure 3b). E-64 (to 20 µM), a cysteine protease inhibitor that inhibits several cathepsins, was used in replicate assays to verify that protease activities (without E-64) were specific to V-CATH. Azocasein was not hydrolyzed in reactions supplemented with E-64 or in mock-, AcEGFP-, or Acp6.9-chiA-infected cells without E-64 supplementation, consistent with protease inhibition by E-64 or lack of *v-cath* in the constructs or prior (Azocoll-based) proteolytic assays [19]. AcMNPV-Rep produced less V-CATH activity than that detected for Acp6.9-chiA/polh-cath. Increased V-CATH activity for Acp6.9-chiA/polh-cath can be attributed to the abundance of *v-cath*-specific mRNAs transcribed from the *polh* promoter (Figure 2a) and V-CATH production (Figure 2b). We expected a larger difference in V-CATH activity between AcMNPV-Rep and Acp6.9-chiA/polh-cath, given the high transcriptional and translational levels in Acp6.9chiA/polh-cath (Figure 2a,b); however, it is also possible that, as reported earlier [25], insoluble proV-CATH detected in immunoblots was not active in this enzyme assay. 

### 3.4. Dual chiA/v-cath Reprogramming Does Not Affect Virus Replication

We previously determined that CHIA overexpression by Acp6.9-chiA did not compromise virus infection kinetics relative to AcEGFP and AcMNPV [19]. There are no chitinous substrates in cell culture that could negatively affect virus cell viability in the presence of overexpressed CHIA. However, Acp6.9-chiA/polh-cath overexpression of V-CATH could have negative cell viability effects and affect the ability of viruses to replicate. Therefore, we compared the kinetics of Acp6.9-chiA/polh-cath infection of cultured cells to that of AcEGFP, AcMNPV-Rep, and Acp6.9-chiA to determine if CHIA and V-CATH enzyme levels were deleterious to the Acp6.9-chiA/polh-cath infection cycle. As with Acp6.9-chiA, there were only slight differences in budded virus production over 72 h among the four viruses (Figure 4). Overexpression of *v-cath* by Acp6.9-chiA/polh-cath may not drastically affect virus replication, since the inactive form of the proV-CATH precursor (Figure 2) may be predominantly produced. It is activated to V-CATH only upon cell death around 48–72 h p.i. 

### 3.5. CHIA and V-CATH Expression in Infected Larvae

We next assessed CHIA and V-CATH production (by immunoblotting and enzyme activity assays) from all viruses following infection of fifth instar *T. ni* larvae. For this, equal amounts of budded virus (5 × 10^4^ TCID_50_ units) were injected into the last proleg of larvae. At various times post-injection, infected larvae were collected and pooled, and chitinase and cathepsin production and enzymatic activities were determined. For immunoblots, we used equal amounts of total protein from the non-sedimented (5000× *g*, 5 min) portion of ground larval tissues at 3 and 4 days p.i. By 5 days p.i., many of the larvae were dead and/or liquefying, making it difficult to collect samples. CHIA was undetectable in tissues of the AcEGFP-infected larvae, indicating that the antibody was specific for the viral chitinase protein. There were similar amounts of CHIA (58 kDa) detected in larval tissues infected with AcMNPV-Rep, Acp6.9-chiA, and Acp6.9-chiA/polh-cath at 3 and 4 days p.i. (Figure 5). There was a notably increased accumulation of CHIA at 4 days p.i. relative to that found at 3 days p.i. for all viruses. We also detected faster-migrating 40/35 kDa bands with the chitinase antibody in Acp6.9-chiA and Acp6.9-chiA/polh-cath samples at day 3. At 4 days p.i., it was detected for all viruses. It is not clear why these lower MW proteins were immunodetected in larval tissues and not cell culture (Figure 2). Presumably, they are derivatives of the viral CHIA, since they were absent from AcEGFP-infected larval samples. 

We also used equal amounts (50 µg) of the same total protein samples from 3 and 4 days p.i. to immunoblot with anti-V-CATH serum. No V-CATH-specific bands were detected from either the AcEGFP or Acp6.9-chiA virus-infected samples, but a non-specific 30 kDa band was detected in all four virus-infected larval protein samples (Figure 5a). There was an increased amount of proV-CATH (35 kDa) detected for the *polh* promoter-expressed *v-cath* by 3 days p.i. relative to that for AcMNPV-Rep. At 3 days p.i., AcMNPV-Rep-infected tissues contained only the inactive 35 kDa proV-CATH enzyme precursor; however, Acp6.9-chiA/polh-cath samples showed a 27 kDa band, corresponding to the mass of the mature V-CATH enzyme. At 4 days p.i., both the AcMNPV-Rep and the Acp6.9-chiA/polh-cath virus-infected larval samples contained mostly the 27 kDa V-CATH band, although some 35 kDa proV-CATH was still detectable in the Acp6.9-chiA/polh-cath protein samples. In accordance with the increased level of proV-CATH detected at 3 days p.i. in the Acp6.9-chiA/polh-cath-infected larvae, relative to that for AcMNPV-Rep at 4 days p.i., there was also considerably more mature 27 kDa V-CATH detected. This indicates that by 4 days p.i., most proV-CATH had been proteolytically processed to the mature V-CATH enzyme form. A non-specific 30 kDa band observed in all virus samples was prominent at 3 days p.i. but slightly detected in the 4-days-p.i. samples. This likely reflects increased proV-CATH/V-CATH abundance (and, therefore, ease of immunodetection over the non-specific 30 kDa band) one day later, as we noted above for differential CHIA levels at 3 and 4 days p.i. 

We also collected hemolymph from infected larvae at 3 days p.i. and immunoblotted proteins. At this time, there was no larval death and very little melanization in larvae infected with any of the viruses. We did not collect hemolymph at 4 days p.i., since several Acp6.9-chiA/polh-cath-infected larvae were melanizing, and this interfered with hemolymph collection. Hemolymph was pooled from at least three individual insects, and equal volumes of hemolymph from each virus group were analyzed. As expected, AcEGFP-infected larval hemolymph did not contain CHIA or proV-CATH/V-CATH. The AcMNPV-Rep-infected larval hemolymph contained a trace amount of CHIA, whereas Acp6.9-chiA and Acp6.9-chiA/polh-cath both had similar amounts of and more CHIA (Figure 5b). 

We also probed for V-CATH in infected larval hemolymph at 3 days p.i. As expected, no cathepsin was detected in hemolymph from the Acp6.9-chiA-infected larvae because the virus does not express *v-cath*. Cathepsin was also not detected in hemolymph from AcMNPV-Rep-infected larvae, but it was detected from that of larvae infected with Acp6.9-chiA/polh-cath (Figure 5b). We also observed that there was a larger proportion of mature 27 kDa V-CATH detected in Acp6.9-chiA/polh-cath-infected hemolymph cells, compared to the level of the 35 kDa inactive proV-CATH progenitor. This suggested that Acp6.9-chiA/polh-cath overexpression of *v-cath* during larval infection resulted in an increase of extracellular V-CATH enzyme capable of prematurely degrading host tissues, as compared to infection with AcMNPV-Rep, and corroborating our earlier report [13]. 

### 3.6. CHIA and V-CATH Enzyme Activity in Infected Larvae

We used total protein derived from infected larvae to perform CHIA and V-CATH enzymatic assays. For this analysis, we collected, pooled, and macerated larvae at 2, 3, and 4 days p.i. The AcEGFP-infected larvae contained background chitinase activity at all timepoints, demonstrating that in total protein (200 µg) derived from the entire larvae, the vast majority of chitinase activity detected was of viral origin (Figure 6a). At days 2 and 3 p.i., there were similar amounts of chitinase activity from AcMNPV-Rep-infected larvae, and these doubled by day 4. There was no significant difference between Acp6.9-chiA and Acp6.9-chiA/polh-cath CHIA activities at any timepoint tested (Figure 6a). However, at all timepoints tested, Acp6.9-chiA and Acp6.9-chiA/polh-cath samples had significantly more chitinase activity than AcMNPV-Rep samples, yet increasing chitinase activity levels were detected each day up to 4 days p.i. for all three (Figure 6a). Therefore, although levels of accumulated chitinase activity differed for virus samples encoding *chiA*, they had similar temporal patterns. 

We also assayed for cathepsin protease activity in infected larvae (400 µg of total protein) using azocasein substrate at days 2, 3, and 4 p.i. (Figure 6b). At 2 days p.i., we detected low levels of protease activity, with a slight but significant increase in Acp6.9-chiA/polh-cath samples. Reactions containing the E-64 inhibitor yielded similar levels of azocasein digestion, indicating that host-derived proteases were responsible for the proteolytic levels observed. Insect protease activity was easily detected in the larval tissues, whereas only background levels were detected in *S. frugiperda*-derived cells (Figure 2; [12,19,25]. At 3 days p.i., there was an increase in protease activity in AcMNPV-Rep and Ac6.9-chiA/polh-cath samples, with a decrease for both at 4 days p.i. However, Acp6.9-chiA/polh-cath V-CATH activity levels were significantly higher than those of AcMNPV-Rep at 3 and 4 days p.i. (Figure 6b).

### 3.7. Dual chiA/v-cath Reprogramming Increases Host Mortality

We monitored the survival times of fifth instar larvae following infection with each of the four viruses by checking every 8 h for larval death, based on lack of response to prodding (Figure 7). Larvae were injected with a large dose (5 × 10^4^ TCID_50_ units) of budded virus. This inoculation route should abrogate any potential confounding effects of differential CHIA/V-CATH levels within occlusion bodies, which would also be an important underlying virulence factor because it might impact oral susceptibilities to each virus. In preliminary experiments, we found similar trends of mortality when 50X lower doses (1 × 10^3^ TCID_50_ units) of each virus were instead injected into fifth instar larvae, but results were more varied (data not shown). Appendix A summarizes the survival data. AcEGFP, which lacks expression of both *chiA* and *v-cath*, was the least pathogenic (median survival time = 120 h). Acp6.9-chiA, which overexpresses *chiA*, but lacks *v-cath* expression, was more pathogenic than AcEGFP (median survival time = 104 h), but less than AcMNPV-Rep (median survival time = 96 h), suggesting that CHIA contributes to pathogenicity directly or indirectly. Since Acp6.9-chiA overexpresses the CHIA enzyme relative to AcMNPV-Rep (Figure 2 and [13,19]), but lacks V-CATH, we do not know if natural AcMNPV-Rep CHIA expression levels in the absence of any V-CATH would result in reduced survival times relative to AcEGFP. However, when *chiA* and *v-cath* were overexpressed by Acp6.9-chiA/polh-cath, it resulted in a further decrease in host survival time (median survival time = 88 h). Acp6.9-chiA/polh-cath killed larvae faster than Acp6.9-chiA or AcMNPV-Rep, indicating that V-CATH may be a more potent virulence factor than CHIA when it is overexpressed. 

We also monitored the progression of *T.ni* larvae pathology following infection with each virus, examining insects every 24 h after injection of budded virus (5 × 10^4^ TCID_50_ units) (Figure 8a). In typical AcMNPV infections of *T. ni* larvae, melanization occurs, and shortly after, larvae succumb to the infection. After death, the larval carcass blackens and liquefies from within, a process facilitated by the concerted enzymatic activities of the viral CHIA and V-CATH [1,18,25,32]. We also determined the cumulative number of live, dead, and liquefied insects in each of the virus-infected groups from 3 to 6 days p.i. (Figure 8b). Larvae infected with any of the viruses had no obvious changes in phenotype before 3 days p.i., including death and color changes (melanization) known to be associated with viral disease, but a few deaths occurred in the AcMNPV-Rep- and Acp6.9-chiA/polh-cath-infected larvae. After 3 days p.i., distinct phenotypes were observed. At 4 days p.i., mock-injected larvae pupated, the AcEGFP-infected larvae were bright green, and melanotic spots developed on the AcMNPV-Rep-, Acp6.9-chiA-, and Acp6.9-chiA/polh-cath-infected larvae. By 4 day p.i., more than 70% of the larvae infected with Acp6.9-chiA/polh-cath had died, and most were considerably more mottled with melanization (if not already completely blackened), compared to AcMNPV-Rep-infected larvae. Notably, several carcasses (about 25%) of Acp6.9-chiA/polh-cath-infected larvae also began to liquefy after 4 days p.i. None of the AcEGFP- nor Acp6.9-chiA-infected larvae melanized or liquefied by 4 days p.i.; the AcEGFP-infected larvae remained bright green, while the Acp6.9-chiA virus-infected larvae became pale green. By 5 days p.i., large proportions (70% or more) of each of the four virus-infected larvae had died. Most (90%) of the AcMNPV-Rep-infected larvae had died and about 50% had blackened and began liquefying. At this point, all of the Acp6.9-chiA/polh-cath-infected larvae had died, and all but one had blackened and began liquefying. By 6 days p.i., nearly all larvae had died, regardless of which virus they were infected with, and all of the AcMNPV-Rep- and Acp6.9-chiA/polh-cath-infected larvae were liquefying. Neither the AcEGFP nor Acp6.9-chiA-infected larvae had melanized considerably or liquefied. The lack of both melanization and liquefaction has previously been attributed to a lack of *v-cath* expression [1,25,32]. 

## 4. Discussion

In this study, we compared the biological activities of viruses with modulated expression of two native viral host tissue-degrading enzymes, CHIA and V-CATH. CHIA is a chitinase related structurally to bacterial chitinases (e.g., chitinase A), but it retains chitinolytic activity across a wide range of pHs [1], unlike orthologous CHIA’s. The V-CATH protease, which likely originated from a host lysosomal cathepsin [20,25,32], has a very broad substrate range and is quite active at neutral pH, in contrast to its cellular counterparts [25]. These enzymes are encoded and produced by most alphabaculoviruses. Betabaculoviruses encode *chiA*, but often not *v-cath,* and instead of *v-cath*, they typically encode matrix metalloproteases (MMPs). None of these enzymes are encoded by delta- or gammabaculoviruses [18]. Their host-tissue degrading ability also makes them candidate virulence factors, because they are host-debilitating enzymes. The native viral expression and enzymatic regulation of both CHIA and V-CATH has evolved to result in an effective viral infection process. CHIA has a KDEL motif that retains the active chitinase enzyme in cells, where there is no chitin substrate, until virus-induced cell death and lysis occurs. The *v-cath* is expressed as an inactive proenzyme (proV-CATH) that, unlike its cellular counterparts, which are typically transported to and spontaneously activated in acidic lysosomes, accumulates in the ER as proV-CATH due to its molecular association with viral CHIA [21]. Terminal host liquefaction is, therefore, accomplished after maximal viral replication and occlusion body formation, only after cells die and release the active CHIA and V-CATH enzymes.

To investigate whether CHIA and V-CATH can be co-opted from naturally occurring viral genes into virulence factors, which might hasten cell death, we attempted to simultaneously modulate both of their expression profiles. In a prior study [19], we successfully used the same methodology to modulate AcMNPV *chiA* transcription and associated CHIA enzyme levels by replacing the native late promoter (i.e., TAAG motif) with other AcMNPV promoters (e.g., late *p6.9* or very late *polh*). We attempted to maintain natural *v-cath* expression profiles in these *chiA*-reprogrammed viruses to limit any potential adverse effects that excessive V-CATH protease activity might impose on the replication cycle. Other reports indicated that *polh* promoter-mediated *v-cath* overexpression did not affect virus viability [13,25], and furthermore, that V-CATH activity was important for release of CHIA (and progeny occlusion bodies) from cells [13]. Prior studies modified expression levels of just the *chiA* or the *v-cath* gene. However, these two host-tissue-degrading enzymes are coordinately regulated during AcMNPV infection of larvae. To initially determine the potential for repurposing these genes as complementary biopesticidal agents, it is critical to assess the pathological and larvicidal effects that result from simultaneous overexpression of both genes by the same virus. Here, we report the results of simultaneously modulated *chiA/v-cath* expression profiles using alternate AcMNPV promoters, inserted into the native *chiA/v-cath* intergenic site, to assess whether simultaneous *chiA*/*v-cath* overexpression can hasten the pathology and mortality associated with baculovirus infection. However, we acknowledge that to truly ascertain whether *chiA*/*v-cath* overexpression could improve the biopesticidal properties (i.e., curb larval feeding and/or larval mortality more rapidly) of a baculovirus, per os infections using the occluded virus form, lethal concentrations and dosage, and quantitative assessment of feeding damage are ultimately required. 

The Acp6.9-chiA/polh-cath, a dual *chiA/v-cath* transcriptionally reprogrammed virus, produced both *chiA* and *v-cath* transcripts in temporal patterns and levels expected from the promoters used vs. AcMNPV-Rep (Figure 2a). Thus, we concluded that it is possible to simultaneously reprogram both *chiA* and *v-cath* transcription from alternate, AcMNPV-derived promoters (*p6.9, polh*) within the short, shared intergenic *chiA/v-cath* promoter region. The *polh* promoter-driven *v-cath* transcription did not impact *p6.9* promoter-mediated *chiA* transcription in the opposite orientation, as evidenced by the abundant transcription of *chiA*. This is in contrast with our previous *p6.9* or *polh* promoter-based *chiA*-reprogrammed viruses, which lacked *v-cath* transcription, even though the native 26 nt *v-cath* promoter sequence, containing the late TAAG *v-cath* mRNA transcription start site [20], was retained [19]. 

We do not know why the adjacent promoters *p6.9* (driving *chiA*) and *polh* (driving *v-cath*) worked together, whereas the native *v-cath* promoter used in our previous study did not function when adjacent to either (oppositely oriented) *polh* or *p6.9* promoters. Perhaps *v-cath* transcription in the single *chiA*-reprogrammed viruses (Acp6.9-chiA/v-cath or Acpolh-chiA/v-cath) [19] was blocked due to truncating the promoter (to 26 nt) or by modifying the *v-cath* promoter sequence context, notions that are supported by a preliminary intergenic 45 bp truncation and substitution study (unpublished, Michael J. Norris, MSc Thesis). The native intergenic *chiA/v-cath* promoter is relatively small (45 bp) and is well conserved in AcMNPV and most other group I alphabaculoviruses and some group II alphabaculoviruses and betabaculoviruses [10,18], and it may be important in regulating coordinated, simultaneously late *chiA* and *v-cath* transcription, so that CHIA and proV-CATH proteins can co-traffic and be co-retained in cells [11]. Perhaps even a small change in the native promoter nucleotide sequence may be sufficient to affect transcription of either gene; yet, upon replacing the intergenic region with alternate promoters, it yields simultaneously altered transcription of both *chiA* and *v-cath*.

Dual *chiA/v-cath* transcriptional reprogramming resulted in increased CHIA and V-CATH enzyme levels within homogenates from virus-infected *S. frugiperda*-derived (SF-21) cells and *T. ni* larvae. We found that Acp6.9-chiA/polh-cath-infected larvae had a shorter survival time (88 h) than either AcMNPV-Rep (96 h) that expresses CHIA and V-CATH or Acp6.9-chiA (104 h) that (over)expresses only CHIA. Larvae infected with AcEGFP, which does not express CHIA nor V-CATH, had the longest survival time (120 h), suggesting that both CHIA and V-CATH naturally influence virus virulence. In addition, these studies show that neither enzyme is essential for virus lethality.

Systemic infections of late instar larvae were initiated by injecting budded virus into the hemolymph (bypassing natural oral inoculation), so we could monitor temporal pathological effects associated with each virus and their CHIA/V-CATH expression profiles. In accordance with the decreasing survival time observed for larval infections with AcEGFP < Acp6.9-chiA < AcMNPV-Rep < Acp6.9-chiA/polh-cath, we noted larval pathology characteristics were commensurate with the levels of CHIA and V-CATH produced by each virus. AcEGFP and Acp6.9-chiA both lacked V-CATH expression/activity, and larvae infected by either virus failed to melanize considerably or liquefy. *T. ni* co-infected with two AcMNPV mutants, one lacking *v-cath* and overexpressing just CHIA(ΔKDEL) and the other (*chiA/v-cath* null) overexpressing Cydia pomonella granulovirus MMP, also failed to liquefy larvae, although partial degradation of internal organs was observed [27]. This highlights the importance of co-expressing these genes for concerted CHIA and V-CATH enzyme function, which has likely been evolutionarily optimized.

It is not clear why CHIA and proV-CATH/V-CATH were easily detectable in hemolymph from larvae infected with Acp6.9-chiA (only CHIA expressed) and Acp6.9-chiA/polh-cath, but not in hemolymph from AcMNPV-Rep-infected larvae, because CHIA should be retained in cells due to its KDEL ER-retention motif. Perhaps the high level of CHIA production, when *chiA* is transcribed under control of the *p6.9* promoter, exceeds the capacity of the cellular ER KDEL receptor function. Similarly, since proV-CATH binds CHIA in the ER of infected cells, where it is usually retained until virus-induced cell lysis occurs [11,17], if the ER CHIA/proV-CATH complex levels exceed the capacity of the ER KDEL receptor, it would lead to premature release of proV-CATH from cells, as shown when the CHIA KDEL was deleted [11]. In addition, we previously reported that *polh* promoter-mediated expression of *v-cath* by a virus lacking *chiA* results in constitutive secretion of a mature-competent proV-CATH, presumably because ER-resident CHIA is not available to bind and retain proV-CATH in cells. It is also possible that the *polh* promoter-mediated overexpression of *v-cath* generates more proV-CATH than can be bound and retained in cells by CHIA, even when *chiA* itself is co-overexpressed with *v-cath*. 

Importantly, while the majority of prior reports describing virulence and other effects that resulted from modifying CHIA and/or V-CATH utilized bacmid-based viruses, here, we describe effects from dual reprogramming of AcMNPV *chiA/v-cath* in their native locus, using just AcMNPV-derived promoter sequences in an otherwise native genomic background. Bacmid-based baculoviruses engineered to overexpress toxic exogenous genes are important for laboratory mortality studies, but such recombinant viruses are typically deemed ineligible for environmental use as biopesticides [33]. Although most alphabaculoviruses and some betabaculoviruses encode and express CHIA and V-CATH, all of the group I alphabaculoviruses (except Anticarsia gemmatalis MNPV) maintain a strictly conserved genomic organization of *chiA* and *v-cath*. Since these genes share a small intergenic region, containing late TAAG promoter motifs responsible for the transcriptional regulation of each gene transcribed from opposite genomic strands, it makes them amenable to the genomic manipulation used herein to reprogram *chiA/v-cath*. Since these genomic modifications do not involve the insertion of any foreign sequences, yet can hasten host pathology and mortality when the virus is delivered by injection into the hemocoel, they provide initial evidence of the potential for developing enhanced baculovirus-based biopesticides. The enhanced larvicidal activities were observed for the Acp6.9-chiA/polh-cath relative to AcMNPV-Rep in acute systemic infections, albeit initiated by injecting budded virus, an unnatural route that is not feasible for biopesticide use. Therefore, it will be important to test if this type of modified virus also has improved mortality responses in oral infections, using occluded virus to determine the differences in the doses necessary for lethality, larval survival times, and larval feeding damage on vegetation. Our current results for this dual *chiA/v-cath* reprogrammed AcMNPV are, nonetheless, promising. 

## Figures and Tables

**Figure 1 viruses-15-00503-f001:**
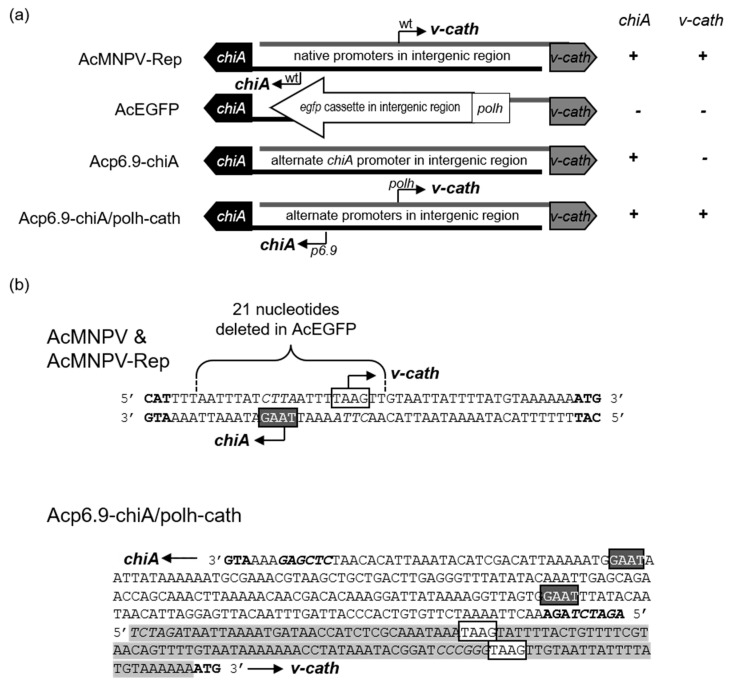
Reprogramming *chiA* and *v-cath* transcription. (**a**) Schematic depicting the method used for switching the AcMNPV *chiA/v-cath* intergenic promoters via an EGFP-selectable virus (AcEGFP) as described before [19]. The small arrows indicate which promoter (wt, *p6.9*, or *polh*) is driving *chiA* or *v-cath* transcription. To the right is a summary of the *chiA* and *v-cath* expression from each virus (+ = expressed, − = not expressed). (**b**) Intergenic *chiA/v-cath* promoter sequence of AcMNPV, AcMNPV-Rep and the dual reprogrammed virus (Acp6.9-chiA/polh-cath). The *chiA* and *v-cath* translation start codons are bolded and restriction enzyme recognition sequences (cloning sites) are italicized. For the dual reprogrammed virus, the sequence of the *p6.9* promoter (driving *chiA*) is shown from 3′-5′ and that of the adjacent *polh* promoter (driving *v-cath,* shaded in grey) are shown from 5′-3′ to reflect the coding strand for each antiparallel ORF. Potential mRNA transcription sites (TAAG) for *chiA* (dark grey) or *v-cath* (white) are boxed. The indicated *chiA* transcription site was mapped for AcMNPV and Acp6.9-chiA [19], but not Acp6.9-chiA/polh-cath. The *v-cath* transcription site was mapped previously [20].

**Figure 2 viruses-15-00503-f002:**
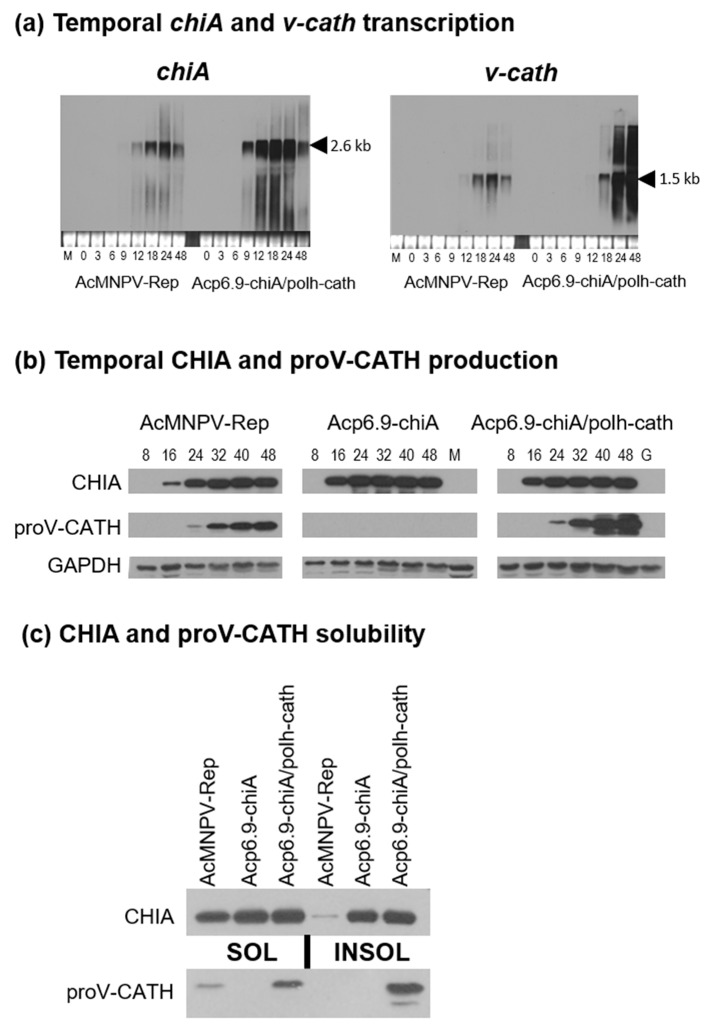
Temporal CHIA and proV-CATH expression by AcMNPV-Rep, Acp6.9-chiA, and Acp6.9-chiA/polh-cath in infected SF-21 cells. (**a**) Northern blots of *chiA* and *v-cath* mRNAs over a time-course of infection of SF-21 cells with AcMNPV-Rep, Acp6.9-chiA, and Acp6.9-chiA/polh-cath. Sizes are based on a high-range RNA ladder (Fermentas). Transcripts were detected using DIG-labeled ssDNA probes as described in [19]. Ethidium bromide-stained rRNA bands indicate RNA equivalency (10 µg/lane). Lane M is an uninfected control sample. (**b**) Temporal CHIA and proV-CATH production (0–48 h p.i.). Total proteins isolated from SF-21 cells infected with the indicated virus were loaded (in equivalent volumes) on gels for protein blots and probed with either anti-BmCHI-h antibody (to detect CHIA), anti-V-CATH antibody (to detect proV-CATH), or anti-GAPDH antibody (to detect host GAPDH) as a loading control. Lane M is protein from uninfected cells as control. Lane G is protein from AcEGFP-infected cells. (**c**) CHIA and proV-CATH solubility. Lysates from infected cells were separated by centrifugation into detergent (0.5% NP-40/1% Triton-X)-soluble (SOL) and -insoluble (INSOL) fractions and proteins were detected with anti-BmCHI-h or anti-V-CATH antibody.

**Figure 3 viruses-15-00503-f003:**
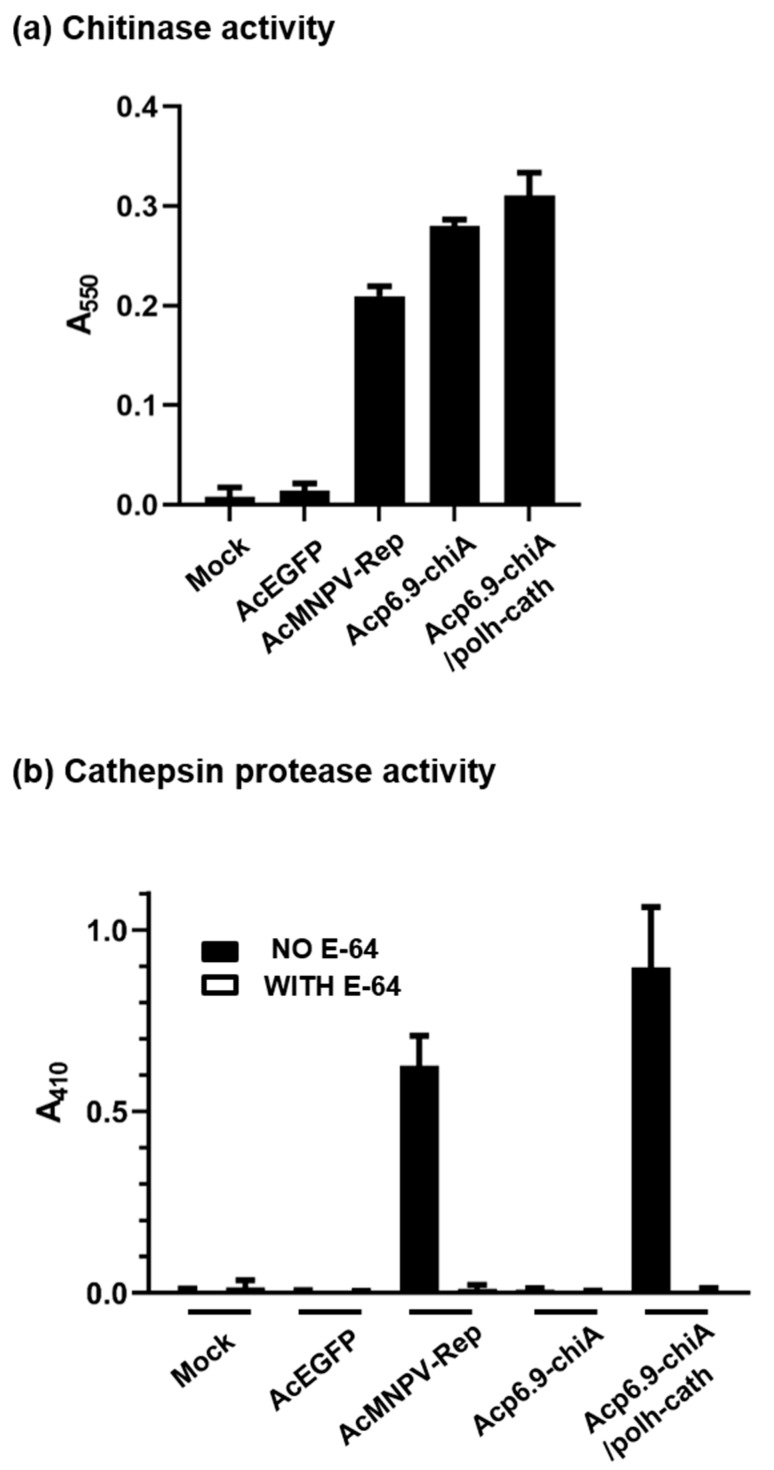
Chitinase and cathepsin (protease) assays. Homogenates from infected SF-21 cells were used to compare viral chitinase (CHIA) and protease (V-CATH) activities at 48 h p.i. with AcMNPV-Rep, Acp6.9-chiA, and Acp6.9-chiA/polh-cath. Mock-infected (Mock) and AcEGFP-infected cells were used as negative controls. (**a**) Chitinase assay. Total protein (50 µg) was used to measure viral CHIA activity. (**b**) Protease assays. Total protein (400 µg) was used to measure V-CATH activity in the presence or absence of 20 µM of the cysteine protease inhibitor E-64.

**Figure 4 viruses-15-00503-f004:**
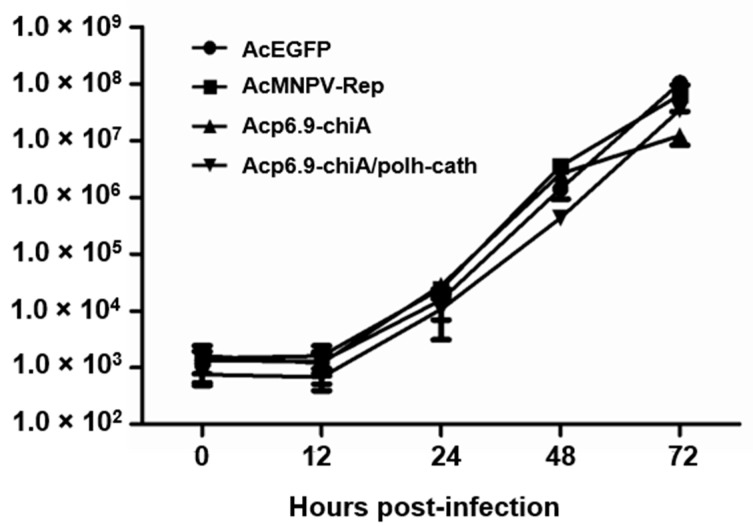
Budded virus production of AcEGFP-, AcMNPV-Rep-, Acp6.9-chiA-, and Acp6.9-chiA/polh-cath-infected cells. SF-21 cells were infected (m.o.i. = 0.1 PFU/cell), and at the indicated timepoints, a small amount (0.2 mL) of supernatant was removed and titrated by end-point dilution. The data points represent the mean budded virus titers from three replicates. Error bars denote standard deviations.

**Figure 5 viruses-15-00503-f005:**
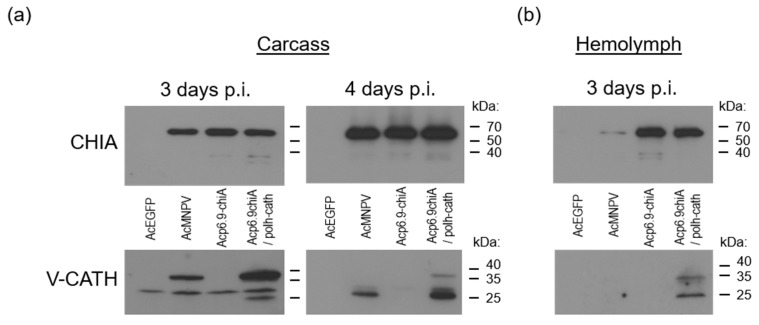
Temporal CHIA and V-CATH protein production from infected *T. ni* larvae. (**a**) Carcass. CHIA (50 µg/lane) and V-CATH/proV-CATH (500 µg/lane) were immunodetected from homogenates of infected larvae collected at 3 and 4 days p.i., as indicated, using anti-BmCHI-h antibody (to detect CHIA) and anti-V-CATH antibody (to detect proV-CATH). (**b**) Hemolymph. CHIA and V-CATH/proV-CATH were immunodetected in pooled (3 larvae each) hemolymph samples extracted at 3 days p.i. The same volume of hemolymph (15 µL) from virus-infected larvae were loaded in each lane. Migration of protein markers is shown on the right in kDa.

**Figure 6 viruses-15-00503-f006:**
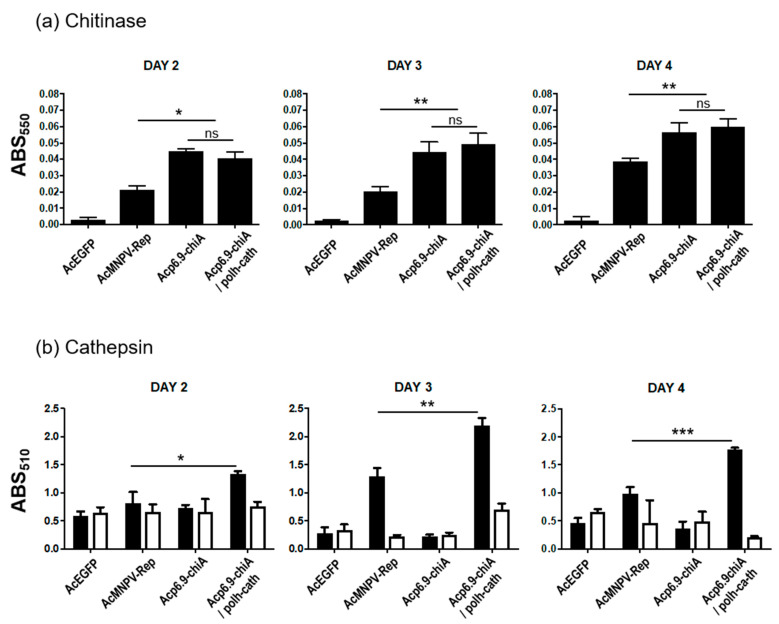
Chitinase and cathepsin (protease) assays from infected *T. ni* larval tissues. Total protein in larval homogenates was used to compare viral chitinase (CHIA) and protease (V-CATH) activities at 3, 4, and 5 days p.i. for AcEGFP, AcMNPV-Rep, Acp6.9-chiA, and Acp6.9-chiA/polh-cath. (**a**) Total protein (200 µg) was used to measure viral chitinase (CHIA) activity. ns, not significant (**b**) Protease assays. Total protein (400 µg) was used to measure V-CATH activity in the presence (white bars) or absence (black bars) of E-64 (20 µM) inhibitor. Statistics were computed only for V-CATH samples without E-64. The indicated *p*-values (*t*-test, 95% CI) were ≤ 0.001 (*), ≤ 0.002 (**), and ≤ 0.003 (***).

**Figure 7 viruses-15-00503-f007:**
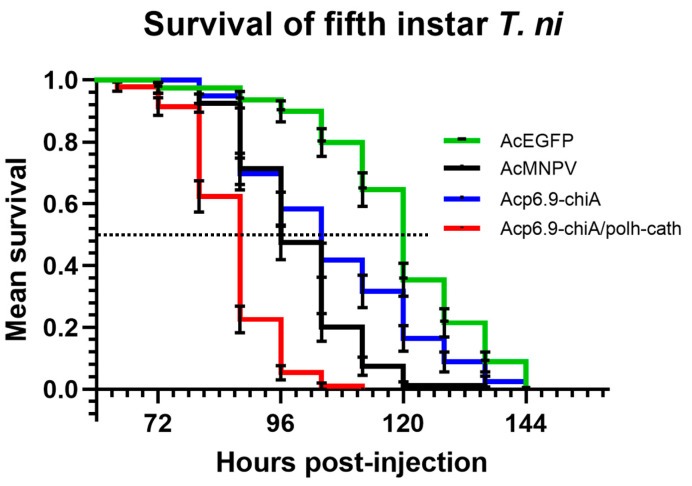
Survival of infected fifth instar *T. ni* larvae. Larvae were injected with 10 µL containing 5 × 10^4^ TCID_50_ units of each virus and were monitored every 8 h for viability at 27 °C. About thirty insects per experiment were injected with each virus and monitored for viability by prodding with a blunt object. The survival curves combine data from three independent experiments. The data were analyzed using the Kaplan–Meier method, and the survival curves were found to be significantly different from each other by Log-rank test (*p* < 0.0001). The dashed horizontal line indicates 50%. Error bars indicate standard error.

**Figure 8 viruses-15-00503-f008:**
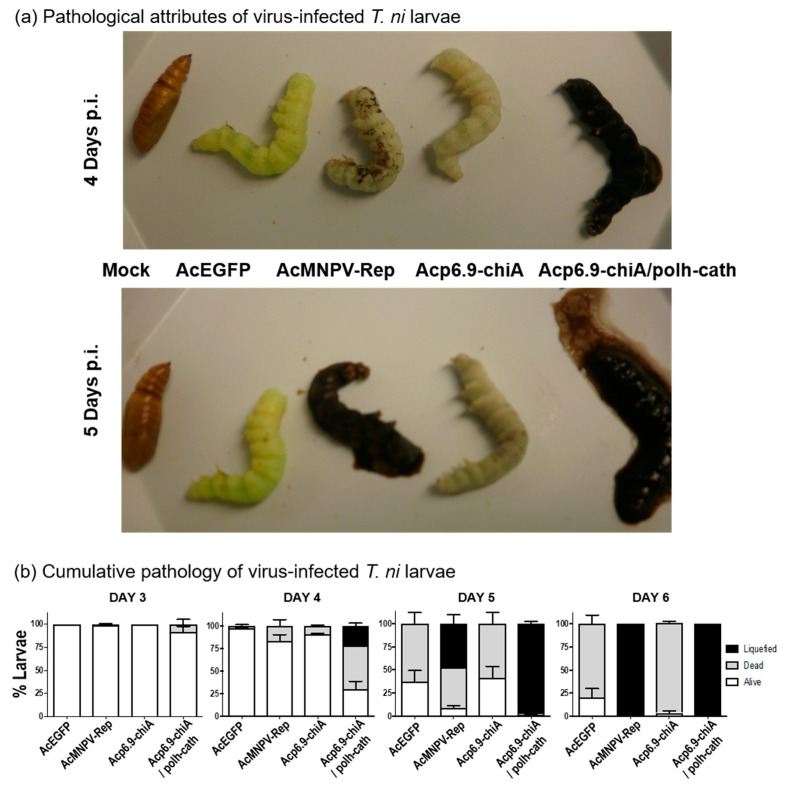
Pathology of infected fifth Instar *T. ni* larvae. Larvae were injected with 10 µL containing 5 × 10^4^ TCID_50_ units of each virus and incubated at 27 °C in individual containers with synthetic diet. They were monitored every 24 h for pathological changes (i.e., live, dead, and liquefied). (**a**) Insects with characteristic phenotypes due to infection with each of the viruses were photographed at 4 and 5 days as indicated. (**b**) The graphed results combine data for 3 to 6 days p.i. for three independent experiments. Error bars indicate standard deviation.

## Data Availability

The data presented in this study are available in [insert article or Appendix A here].

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
