# Peer review of "Transcriptional Reprogramming of Autographa Californica Multiple Nucleopolyhedrovirus Chitinase and Cathepsin Genes Enhances Virulence"

_viruses, 2023, doi:10.3390/v15020503_

Round 1
Reviewer 1 Report
This is a report about “Transcriptional reprogramming of Autographa californica multiple nucleopolyhedrovirus chitinase and cathepsin genes enhances virulence” by Hodgson et al. The work is very interesting and clear; conclusions are properly taken from the produced data; material and methos are consistent; and results are sufficiently robust to be published in a high impact factor journal like Viruses. I have only very specific concerns that must be addressed by the authors prior publication. Authors justify the paper by the enhancing AcMNPV virulence by manipulating virus genome with its own sequences (i.e. promoters) to avoid considering it a genetically modified virus. I agree with this though and I think this should be clearly addressed in the discussion section. However, methods do not support their hypothesis. Authors did not consider using the natural route for insect infection to make sure that the claimed ‘enhanced virulence’ would be maintained by oral infection route. I think findings themselves are interesting enough to be published as basic research and there is no need to justifying the paper by the use of the virus as biological control agent. The authors must consider removing this focus from the manuscript or they should provide an oral bioassay that gets closer to the natural route of infection. An important point, the reprogramming is not only at transcriptional level, but also at translational and enzymatical levels. I think authors should consider changing the title from ‘transcriptional reprogramming’ to “reprogramming of AcMNPV chitinase and cathepsin expression’.
Some specific comments:
Line 31: provide a reference for “auxiliary genes to virus replication”. What do the authors mean with ‘virus replication’?
Figure 1A: the figure is confusing. Rather than ‘alternate ChiA promoter in intergenic region’, please specify the promoters and their direction in the scheme.
Figure 1B: please, add some highlights (colors, underlined) to separate p6.9 promoter from polh promoter sequence.
Line 65: replace ‘control’ to ‘parental’.
Line 66: authors must consider using ‘chiA and v-cath sequence-containing RNAs’ rather than mRNA. Baculoviruses can transcribe extended and long mRNAs that may contain chiA and v-chat sequence downstream from the real coding sequence (which is closer to the 5’-UTR). That mRNA is not able to express chiA and v-cath although it contains chiA or v-cath sequences.
Author Response
Reviewer 1:
(reviewer general comments)
This is a report about “Transcriptional reprogramming of Autographa californica multiple nucleopolyhedrovirus chitinase and cathepsin genes enhances virulence” by Hodgson et al. The work is very interesting and clear; conclusions are properly taken from the produced data; material and methos are consistent; and results are sufficiently robust to be published in a high impact factor journal like Viruses. I have only very specific concerns that must be addressed by the authors prior publication. Authors justify the paper by the enhancing AcMNPV virulence by manipulating virus genome with its own sequences (i.e. promoters) to avoid considering it a genetically modified virus. I agree with this though and I think this should be clearly addressed in the discussion section. However, methods do not support their hypothesis. Authors did not consider using the natural route for insect infection to make sure that the claimed ‘enhanced virulence’ would be maintained by oral infection route. I think findings themselves are interesting enough to be published as basic research and there is no need to justifying the paper by the use of the virus as biological control agent. The authors must consider removing this focus from the manuscript or they should provide an oral bioassay that gets closer to the natural route of infection. An important point, the reprogramming is not only at transcriptional level, but also at translational and enzymatical levels. I think authors should consider changing the title from ‘transcriptional reprogramming’ to “reprogramming of AcMNPV chitinase and cathepsin expression’.
This reviewer had some general comments and suggestions to improve, and more accurately describe our findings and their importance of using this methodology for biopesticides.
This reviewer (and reviewer #2) had concerns about regarding the chiA/v-cath overexpressing virus as a more virulent baculovirus biopesticide, because we injected virus and did not perform oral bioassays with the occluded form of the virus (the natural route of infection). We agree that the oral bioassays are critical to assessing the real potential of the chiA/v-cath overexpressing virus as a biopesticide, which would require oral exposure in field use.
We realize the concerns of describing the enhanced larvicidal effects of the dual chiA/v-cath reprogrammed virus in terms of its potential for producing more effective orally acquired baculovirus-based biopesticides. However, we consider this current study as preliminary evidence of how simultaneously altering chiA and v-cath expression impacts larval mortality.
While we contend this study does show that there is potential for dual chiA/v-cath overexpression to increase “virulence relative to the wt and other viruses”, we do acknowledge the disconnect in directly attributing chiA and v-cath over-expression to “an oral biopesticide with enhanced virulence”. To address this, we have overtly stated this shortcoming in the methods used for this study (injecting high virus doses) in the discussion.
We modified our current statement(s) in the original discussion (lines 558-563):
“Therefore, to determine the full potential for repurposing these genes as complementary biopesticidal agents it is critical to assess pathological and larvicidal effects that result from simultaneous overexpression of both genes by the same virus. Here we report results of simultaneously modulated chiA/v-cath expression profiles using alternate AcMNPV promoters, inserted into the native chiA/v-cath intergenic site.“
To instead read:
“To initially determine the potential for repurposing these genes as complementary biopesticidal agents, it is critical to assess pathological and larvicidal effects that result from simultaneous overexpression of both genes by the same virus. Here, we report results of simultaneously modulated chiA/v-cath expression profiles using alternate AcMNPV promoters, inserted into the native chiA/v-cath intergenic site, for assessing whether simultaneous chiA/v-cath overexpression can hasten the pathology and mortality associated with baculovirus infection. However, we acknowledge that to truly ascertain whether chiA/v-cath overexpression could improve biopesticidal properties (i.e., curb larval feeding and/or larval mortality more rapidly) of a baculovirus, per os infections using the occluded virus form, lethal concentrations and dosage, and quantitative assessment of feeding damage are ultimately required.“
And we also altered the original discussion statements (line 638-643):
“Since these genomic modifications do not involve the insertion of any foreign sequences, as shown in this study, and increase host mortality, they provide a tool to manufacture enhanced baculovirus-based biopesticides. The enhanced larvicidal activities were observed for Acp6.9-chiA/polh-cath relative to AcMNPV-Rep in acute systemic infections (initiated by injecting budded virus). It will be important to test if these viruses have improved mortality responses in oral infections with occluded virus and doses necessary for lethality.”
To instead read:
“Since these genomic modifications do not involve the insertion of any foreign sequences, yet can hasten host pathology and mortality when the virus is delivered by injection into the hemocoel, they provide initial evidence of the potential for developing enhanced baculovirus-based biopesticides. The enhanced larvicidal activities were observed for Acp6.9-chiA/polh-cath relative to AcMNPV-Rep in acute systemic infections, albeit initiated by injecting budded virus, an unnatural route that is not feasible for biopesticide use. Therefore, it will be important to test if this type of modified virus also has improved mortality responses in oral infections using occluded virus to determine differences in doses necessary for lethality, larval survival times, and larval feeding damage.”
This reviewer also suggested considering rewording the nature of chiA/v-cath “transcriptional reprogramming” to just “reprogramming” in the title so it also encompasses the effects at the CHIA/proV-CATH translation and enzymatic levels. While this is a minor comment, in our methods we indeed only granted the ability of the virus to have altered transcription of these genes since we simply altered their promoters. We cannot predict the influence of how the infected cell will deal with the differential level of chiA/v-cath RNA transcripts relative to those from the native virus chiA/v-cath transcription levels. We regard the increased translation and enzymatic of these two overexpressed proteins to be the result of transcriptional reprogramming, and thus feel the designation of “transcriptional reprogramming” is the most accurate and specific description based on our construction of recombinant viruses.
Response to Specific Comments:
Reviewer Comment-Line 31: provide a reference for “auxiliary genes to virus replication”. What do the authors mean with ‘virus replication’?
The reviewer asked for a reference attesting the auxiliary nature of chiA and v-cath, and a clarification of what we mean by “virus replication”. They specifically wanted referencing and elaboration of the statement on line 31:
“The native baculovirus chiA and v-cath genes are both auxiliary genes to virus replication and are not conserved in all baculoviruses.”
We have added references to support the designation of chiA and v-cath as auxiliary genes, and have modified the statement on line 31 to better elaborate on the meaning of replication:
“The native baculovirus chiA and v-cath genes are not conserved in all baculoviruses, and are each considered an auxiliary gene [1, 2] for per os infection and the cellular production of budded and occluded virions.”
Reviewer Comment-Figure 1A: the figure is confusing. Rather than ‘alternate ChiA promoter in intergenic region’, please specify the promoters and their direction in the scheme.
Reviewer Comment-Figure 1B: please, add some highlights (colors, underlined) to separate p6.9 promoter from polh promoter sequence
The reviewer found Figure 1 confusing, and asked for specific indicators of promoters for each gene in the schematic in 1B as well as adding some shading as a way to identify the p6.9 and polh promoter sequences in 1B.
We have modified Figure 1A / B to incorporate this information so it is more clear what the promoters are in the schematics (we added directional arrows and named the promoter and transcript produced) and the polh promoter sequence was shaded in grey to distinguish it from the p6.9 promoter sequence in 1B. We also adjusted the figure legend to reflect these changes.
Reviewer Comment-Line 65: replace ‘control’ to ‘parental’.
The reviewer suggested changing “control” to “parental” in describing the AcMNPV virus we used for comparisons to the other viruses on line 65:
“Compared to the control AcMNPV virus, Acp6.9-chiA/polh-cath has differential transcription (mRNA) patterns of both chiA and v-cath mRNA and protein production levels of CHIA and proV-CATH.”
We have modified the statement to more accurately name the virus used for comparison to the other, chiA/v-cath reprogrammed viruses. However, the “control” virus in this case is not in fact a parental virus. As we describe in the methods, the control virus was derived from AcEGFP like all the other derivative viruses in this study – but it was engineered to repair the native 45 bp intergenic AcMNPV chiA/v-cath promoter sequences.
Therefore, we adjusted the statement to instead read:
“Compared to the control virus (AcMNPV-Rep), Acp6.9-chiA/polh-cath has differential transcription (RNA) patterns of both chiA and v-cath RNA and protein production levels of CHIA and proV-CATH.”
Reviewer Comment-Line 66: authors must consider using ‘chiA and v-cath sequence-containing RNAs’ rather than mRNA. Baculoviruses can transcribe extended and long mRNAs that may contain chiA and v-chat sequence downstream from the real coding sequence (which is closer to the 5’-UTR). That mRNA is not able to express chiA and v-cath although it contains chiA or v-cath sequences.
The reviewer also suggested the authors to reconsider re-naming (chiA and v-cath) “mRNA” to “chiA and v-cath sequence-containing RNAs” to indicate that not all RNAs containing chiA and v-cath sequences are “mRNA” for translation of either into a protein.
We do recognize that late baculovirus RNAs are often produced as overlapping but not functional polycistonic mRNAs. However, in the Northern blots (Fig 2a) we assessed to formulate this conclusion/statement on mRNAs, we are specifically describing (and in figure 2a pointing to) the major band in the blots for each RNA, so we feel it unnecessary to elaborate on this. Using the reviewers designation for mRNA, will confuse the reader and may be interpreted as other gene(s) having the effects we observed. Since we did not actually determine the transcriptional start site of the major RNA species (e.g., by RACE), we have opted to remove the “m” from “mRNA” (in 2 instances) to make the statement more accurate as to the nature of the chiA/v-cath transcripts.
Reviewer 2 Report
Hodgson and co-authors described an approach to modify baculoviruses to increase their pathogenicity in insect larvae without introducing foreign genes. The idea of over expressing the chitinase (CHIA) and cathepsin (V-CATH) enzymes to generate this effect is not totally new due to the previous knowledge of the role of these proteins in the pathogenesis of baculovirus-infected insects. However, in my knowledge, this is the first time that the over expression of both proteins was explored. The study is quite conclusive with respect to the effects found in injected larvae with high virus doses but lack of the demonstration of the effects that could be found in naturally per os infected larvae, which should be the route of infection when these genetically modified baculoviruses could be applied to control pests. The fact that authors described the most significant effects of Acp6.9-chiA/polh-cath baculovirus when high but not with low virus doses could limit the applicability of these baculoviruses in real situations..
As mentioned in the discussion by the authors, studies to determine larvicidal activity and the potential to curb larval feeding damage upon per os infection with Acp6.9-chiA/polh-cath relative to that for wild-type baculoviruses is of great relevance. Experiments with occluded viruses should be necessary for final conclusions about the applicability of this work. In addition, other combinations of promoters to express both enzymes could also influence the results in insect pathogenicity, especially if the promoters used accelerate the expression of enzymes instead to express them late after infection. Early expression could be even more relevant than over expression.
Additional comments:
The number of insects used in the experiments are relatively low
The figure 5 is confusing with respect to the molecular weights indicated for the proteins detected and there is not a correspondence with the text in results, which indicates different protein mobilities.
Author Response
Reviewer 2:
(reviewer general comments)
Hodgson and co-authors described an approach to modify baculoviruses to increase their pathogenicity in insect larvae without introducing foreign genes. The idea of over expressing the chitinase (CHIA) and cathepsin (V-CATH) enzymes to generate this effect is not totally new due to the previous knowledge of the role of these proteins in the pathogenesis of baculovirus-infected insects. However, in my knowledge, this is the first time that the over expression of both proteins was explored. The study is quite conclusive with respect to the effects found in injected larvae with high virus doses but lack of the demonstration of the effects that could be found in naturally per os infected larvae, which should be the route of infection when these genetically modified baculoviruses could be applied to control pests. The fact that authors described the most significant effects of Acp6.9-chiA/polh-cath baculovirus when high but not with low virus doses could limit the applicability of these baculoviruses in real situations..
As mentioned in the discussion by the authors, studies to determine larvicidal activity and the potential to curb larval feeding damage upon per os infection with Acp6.9-chiA/polh-cath relative to that for wild-type baculoviruses is of great relevance. Experiments with occluded viruses should be necessary for final conclusions about the applicability of this work. In addition, other combinations of promoters to express both enzymes could also influence the results in insect pathogenicity, especially if the promoters used accelerate the expression of enzymes instead to express them late after infection. Early expression could be even more relevant than over expression.
This reviewer provided extended insight on how this study could be improved or at least furthered, for example by using different combinations of promoters (i.e., early) to alter chiA/v-cath expression to produce potentially more profound effects on larval mortality. We acknowledge this would be an interesting avenue for further work but outside the scope of this study, studying one temporal phase. We also note that changing the timing of late genes to early times when the virus has not fully replicated, may not have effects that would help further define the function of v-cath and chiA.
This reviewer, like reviewer #2, also suggested that the enhanced larvicidal effects we report based on injection of high budded virus doses into hemolymph, rather than injected lower doses or via per os infection with the occluded virus (the natural infection route) is not the ideal experimental set-up for assessing for potential enhanced biopesticide activity.
As stated above in response to reviewer 1 and reiterated here, we realize the concern for describing the enhanced larvicidal effects of the dual chiA/v-cath reprogrammed virus in terms of its potential for producing more effective orally acquired baculovirus-based biopesticides. However, we consider this current study as preliminary evidence of how simultaneously altering chiA and v-cath expression impacts larval mortality.
Reviewer Comment-The study is quite conclusive with respect to the effects found in injected larvae with high virus doses but lack of the demonstration of the effects that could be found in naturally per os infected larvae, which should be the route of infection when these genetically modified baculoviruses could be applied to control pests. The fact that authors described the most significant effects of Acp6.9-chiA/polh-cath baculovirus when high but not with low virus doses could limit the applicability of these baculoviruses in real situations
While we contend this study does show that there is potential for dual chiA/v-cath overexpression to increase “virulence relative to the wt and other viruses”, we do acknowledge the disconnect in directly attributing chiA and v-cath over-expression to “a biopesticide with enhanced virulence”. To address this, we have overtly stated this shortcoming in the methods used for this study (injecting high virus doses) in the discussion.
We modified our current statement(s) in the original discussion (lines 558-563):
“Therefore, to determine the full potential for repurposing these genes as complementary biopesticidal agents it is critical to assess pathological and larvicidal effects that result from simultaneous overexpression of both genes by the same virus. Here we report results of simultaneously modulated chiA/v-cath expression profiles using alternate AcMNPV promoters, inserted into the native chiA/v-cath intergenic site.“
To instead read:
“To initially determine the potential for repurposing these genes as complementary biopesticidal agents it is critical to assess pathological and larvicidal effects that result from simultaneous overexpression of both genes by the same virus. Here we report preliminary results of simultaneously modulated chiA/v-cath expression profiles using alternate AcMNPV promoters, inserted into the native chiA/v-cath intergenic site, for assessing whether simultaneous chiA/v-cath overexpression can hasten the pathology and mortality associated with baculovirus infection. However, we acknowledge that to truly ascertain whether chiA/v-cath overexpression could improve biopesticidal properties (i.e. curb larval feeding and/or larval mortality more rapidly) of a baculovirus, per os infections using the occluded virus form, lethal concentrations and doses, and quantitative assessment of feeding damage are ultimately required.“
And we also altered the original discussion statements (lines 638-643):
“Since these genomic modifications do not involve the insertion of any foreign sequences, as shown in this study, and increase host mortality, they provide a tool to manufacture enhanced baculovirus-based biopesticides. The enhanced larvicidal activities were observed for Acp6.9-chiA/polh-cath relative to AcMNPV-Rep in acute systemic infections (initiated by injecting budded virus). It will be important to test if these viruses have improved mortality responses in oral infections with occluded virus and doses necessary for lethality.”
To instead read:
“Since these genomic modifications do not involve the insertion of any foreign sequences, yet can hasten host pathology and mortality, when the virus is delivered by injection into the hemocoel, they provide initial evidence of the potential for developing enhanced baculovirus-based biopesticides. The enhanced larvicidal activities were observed for Acp6.9-chiA/polh-cath relative to AcMNPV-Rep in acute systemic infections, albeit initiated by injecting budded virus, an unnatural route that is not feasible for biopesticide use. Therefore, it will be important to test if this type of modified virus also has improved mortality responses in oral infections using occluded virus to determine any differences in: doses necessary for lethality, larval survival times and larval feeding damage.”
This reviewer had also commented that the number of insects assessed was relatively low.
Reviewer Comment-The number of insects used in the experiments are relatively low
Although more insects would provide additional corroboration of the results, using 30 insects per assay provides enough replicates to make a conclusion. In addition, we collected data at different times post infection, providing additional numbers to validate the data and a trend. In addition, we conducted three independent assays, using 30 insects for each virus and in each iteration for the pathology/mortality assessments and the data from each iteration were in close agreement. With this, we are confident that our experiment reflects the conclusions.
Reviewer Comment-The figure 5 is confusing with respect to the molecular weights indicated for the proteins detected and there is not a correspondence with the text in results, which indicates different protein mobilities.
We addressed this situation in the CHIA blots by better estimating the sizes of the “unexpected additional bands for AcMNPV-Rep and Acp6.9-chiA/polh-cath” by modifying our original statement (lines 380-382):
“We also detected a faster migrating 35 kDa band with the chitinase antibody in Acp6.9-chiA and Acp6.9-chiA/polh-cath samples at day 3. At 4 days p.i., it was detected for all viruses.”
To instead read:
“We also detected faster migrating 40/35 kDa bands with the chitinase antibody in Acp6.9-chiA and Acp6.9-chiA/polh-cath samples at day 3.”
To address this situation for the V-CATH immunoblots, we re-evaluated the positions of the protein standards marked on the on the blots, and adjusted the demarcations for the 25 and 35 kDa bands in Figure 5 to better reflect the actual positions. We acknowledge that the 25 kDa protein standard band position is coincident with the migration of the 27 kDa V-CATH band, but this is expected in these type of gels.
Reviewer 3 Report
Previous studies by this group has only changed the expression profile of chiA. The current manuscript is an extension of the previous paper. The authors changed the expression profiles of chiA and v-cath by introducing p6.9 and polh promoters. The reprogrammed virus demonstrates better insecticidal toxicity, which provided an effective strategy for biological control of insects.
Line 159: ‘The anti-V-CATH was used at 1:1,000 dilution’. If possible, please provide more background information about this antibody (monoclonal or polyclonal antibody?).
Figure 5a. panel V-CATH. How to explain the 30kD bands which were detected in all four virus-infected larval at 3 dpi but not at 4 dpi.
Author Response
Reviewer 3:
(reviewer general comments)
Previous studies by this group has only changed the expression profile of chiA. The current manuscript is an extension of the previous paper. The authors changed the expression profiles of chiA and v-cath by introducing p6.9 and polh promoters. The reprogrammed virus demonstrates better insecticidal toxicity, which provided an effective strategy for biological control of insects.
We appreciate the reviewer understanding the additional information that this study provides on previous work.
Reviewer Comment-Line 159: ‘The anti-V-CATH was used at 1:1,000 dilution’. If possible, please provide more background information about this antibody (monoclonal or polyclonal antibody?).
We used this antibody, obtained from and produced by Jeff Slack, in our prior studies, and had referred to this in the methods. We originally included references so the reader can obtain more details on antibodies used in this study.
However, we added some additional information about the nature of this antibody in the methods, line 159-160:
Original description:
“The anti-V-CATH was used at 1:1,000 dilution and the anti-BmCHI-h was used at 1:50,000 dilution. “
Amended description:
“The rabbit polyclonal anti-V-CATH, which recognizes both proV-CATH and V-CATH, was used at 1:1,000 dilution and the anti-BmCHI-h was used at 1:50,000 dilution. “
Reviewer Comment-Figure 5a. panel V-CATH. How to explain the 30kD bands which were detected in all four virus-infected larval at 3 dpi but not at 4 dpi.
The reviewer is correct that one can see faint bands in this position (albeit not for AcEGFP) in the 4 day samples.
To address this, we added this statement in the results discussing these blots.
“A non-specific 30 kDa band observed in all virus samples was prominent at 3 days p.i., but slightly detected in the 4 days p.i. samples. This likely reflects increased proV-CATH/V-CATH abundance (and therefore ease of immunodetection over the non-specific 30 kDa band) one day later, as we noted above for differential CHIA levels on days 3 and 4 p.i.. “